# Application of Highly Immunocompromised Mice for the Establishment of Patient-Derived Xenograft (PDX) Models

**DOI:** 10.3390/cells8080889

**Published:** 2019-08-13

**Authors:** Seiji Okada, Kulthida Vaeteewoottacharn, Ryusho Kariya

**Affiliations:** 1Division of Hematopoiesis, Joint Research Center for Human Retrovirus Infection, Kumamoto University, Kumamoto 860-0811, Japan; 2Graduate School of Medical Sciences, Kumamoto University, Kumamoto 860-0811, Japan; 3Department of Biochemistry, Khon Kaen University, Khon Kaen 40002, Thailand; 4Cholangiocarcinoma Research Institute, Khon Kaen University, Khon Kaen 40002, Thailand

**Keywords:** patient-derived xenograft, immunocompromised mice, precision medicine, drug screening, cancer, cell line, cancer immunotherapy, humanized mice

## Abstract

Patient-derived xenograft (PDX) models are created by engraftment of patient tumor tissues into immunocompetent mice. Since a PDX model retains the characteristics of the primary patient tumor including gene expression profiles and drug responses, it has become the most reliable in vivo human cancer model. The engraftment rate increases with the introduction of Non-obese diabetic Severe combined immunodeficiency (NOD/SCID)-based immunocompromised mice, especially the NK-deficient NOD strains NOD/SCID/interleukin-2 receptor gamma chain(IL2Rγ)^null (^NOG/NSG) and NOD/SCID/Jak3(Janus kinase 3)^null^ (NOJ). Success rates differ with tumor origin: gastrointestinal tumors acquire a higher engraftment rate, while the rate is lower for breast cancers. Subcutaneous transplantation is the most popular method to establish PDX, but some tumors require specific environments, e.g., orthotropic or renal capsule transplantation. Human hormone treatment is necessary to establish hormone-dependent cancers such as prostate and breast cancers. PDX mice with human hematopoietic and immune systems (humanized PDX) are powerful tools for the analysis of tumor–immune system interaction and evaluation of immunotherapy response. A PDX biobank equipped with patients’ clinical data, gene-expression patterns, mutational statuses, tumor tissue architects, and drug responsiveness will be an authoritative resource for developing specific tumor biomarkers for chemotherapeutic predictions, creating individualized therapy, and establishing precise cancer medicine.

## 1. Introduction

Preclinical studies using animal models are essential for drug development. Fewer than 10% of candidate drugs are approved for the market, even if preclinical trials are successful [1]. This figure is lower for oncology drugs, at approximately 5% [2]. One possible reason is the lack of appropriate human cancer models. Mouse tumors and human-cell-line-transplanted animal models are not always representative of human cancer pathologies, contributing to distinct drug responses [3]. Mice and humans are considerably different [4], and human cancer cell lines somehow lose their original tumor characteristics when transplanted [5]. Accordingly, the National Cancer Institute (NCI, MD, USA) recently decided to replace the NCI-60, a panel of 60 human cell lines, with patient-derived xenografts (PDXs) for drug screening [3]. The PDXs are established by direct engraftment of a patient tumor into an immunocompromised mouse, maintaining the tumor growth in vivo. This has become an essential tool for preclinical and translational research, particularly for investigations of tumor pathology and for chemotherapeutic drug development. With the introduction of highly immunocompromised mice as recipients, PDX use is now widespread and is becoming a standard “Avatar” model for human cancer research.

## 2. Establishment of Immunocompromised Mice

### 2.1. Nude Mice

In 1962, the first known immunocompromised mice were discovered by Grist (Ruchill Hospital, Glasgow, UK). The “nude” nickname was given because they lacked body fur. Flanagan [6] showed that nude mice also lacked thymus and T lymphocytes; as a result, the adaptive immune responses, including T cell-mediated immune responses and antibody formation that require helper T cells, are defective in nude mice. Since then, nude mice have been used as recipients for human tumor xenografts. However, the intact (or rather activated) innate immunity in nude mice limits the options for human cancer transplantation [7]. In addition, nude mice show leakage of T cells with age [8].

### 2.2. Severe Combined Immunodeficient Mice

In 1983, Bosma [9] (Fox Chase Cancer Institute, PA, USA) first described the severe combined immunodeficient (SCID) mice that lack both functional T and B lymphocytes. The maturation deficiencies of B and T lymphocytes in SCID mice are due to the deletion of Prkdc (protein kinase, DNA activated, catalytic polypeptide: DNA-PKCs) and the absence of variable (V)-diveresity (D)- joining (J) recombination (V(D)J recombination). The SCID mice were first used as recipients of human hematopoietic stem cells (HSCs) and peripheral blood mononuclear cell (PBMC) transplantation [10,11]. The engraftment efficiency of human tumors is higher in SCID mice compared to nude mice [12]. However, the transplantation efficiencies of human blood cells and tumor cells are not as high as expected, as the remnant natural killer (NK) cells prevent homing and maintenance of human cells. To overcome the effects of NK cells, SCID/Beige mice were established by crossbreeding SCID mice and Beige mice [13]. In addition to the T and B deficiency of SCID mice, the SCID/Beige mice displayed severely reduced NK cell functioning along with the phagocytosis of Beige mice [14]. The uptake rate of human tumor cells increased in SCID/Beige mice compared with SCID mice, as expected [14], but the engraftment rate of human HSCs was not noticeably increased [15].

### 2.3. Non-Obese Diabetic/SCID Mice and NOD/SCID-Based Immunocompromised Mice

Non-obese diabetic (NOD) mice, discovered in 1980 by Makino (Shionogi Co. Osaka, Japan), have diabetes mellitus caused by infiltration of and pancreatic islet destruction by T lymphocytes [16]. Later, it was discovered that NOD mice acquire multiple immune abnormalities including loss of complement and impaired NK, macrophage, and dendritic cell functions [17]. The NOD/SCID mice were established by crossbreeding NOD and SCID mice. The NOD/SCID mice do not develop diabetes because they lose functional T lymphocytes. Moreover, multiple defects in innate and adaptive immunity are seen in these mice, making them better recipients for human hematopoietic stem cell and human solid tumor transplantation [18]. However, there are some residual NK activities in NOD/SCID mice, and several attempts were made to eliminate or suppress these and to improve transplantation efficiency. These attempts included the use of anti-interleukin (IL)-2 receptor antibody or anti-ganglio-N-tetraosylceramide (asialoGM1) antibody and crossbreeding with β2 macroglobulin- or perforin-deficient mice. The common γ chain (γc, CD132), also known as IL-2 receptor subunit gamma (IL2Rγ), is a cytokine receptor sub-unit. It is common to the receptor complexes for six different interleukin receptors, IL-2, IL-4, IL-7, IL-9, IL-15, and IL-21, which are critical for lymphocyte and NK cell development [19]. The IL2Rγ interacts with the Janus kinase 3 (Jak3) non-receptor-type tyrosine kinase for signal transduction. Therefore, IL2Rγ- and Jak3-deficient mice show common phenotypes such as NK deficiency and reduction of T and B lymphocytes [20]. Thus, NOD/SCID mice with complete loss of NK cells were established by crossbreeding with IL-2 receptor γ-deficient mice (NOD/SCID/IL2Rγ^nul^: NOG [21], NOD/SCID/IL2Rγ^nul^: NSG [22]) or Jak3-deficient mice (NOD/SCID/Jak3^null^: NOJ [23]) (Table 1). NOG mice have a NOD/ShiJic-Prkdc^scid^ background with partial deficiency of IL2Rγ [21], whereas NSG mice have a NOD/ShiSzJ-Prkdc^scid^ background with complete deficiency of IL2Rγ [10]. The NSG mice acquire higher engraftment capacity of cord-blood-derived CD34^+^ cells [24] and higher body weights [25], but these differences do not appear to affect the PDX transplantation efficiency [25]. Moreover, signal regulatory protein alpha (Sirpα) polymorphism in NOD strains provides superior opportunities for human cell engraftment because SIRPα interacts with human CD47 [26] and suppresses macrophage-mediated phagocytosis, or contributes to the so-called “don’t eat me” signal (Figure 1) [27,28].

### 2.4. BALB/c Background Immunocompromised Mice

The BALB/c mice also have Sirpα polymorphisms that acquire binding affinity to human CD47. Hence, BALB/c strain immunocompromised mice, such as BALB/c Rag-2^null^/IL2Rγ^null^ (BRG) [29] and Rag-2^null^/Jak3^null^ (BRJ) mice [30] have lower macrophage-mediated phagocytosis of human cells, and might be useful recipients for human cell and tissue transplantations [31,32]. Other mice such as C57/BL6 have lower affinity for engrafting human normal and malignant cells [30,33]. Even though the mutations in SCID mice are useful for T and B cell elimination, there are several disadvantages to SCID mice, such as high susceptibility to irradiation or drugs, and leakage of T lymphocytes. (Recombination activating gene-1/2 (Rag-1/Rag-2) knockout mice are generated to 1) solve these problems and 2) eliminate mature lymphocytes (Table 2) [29,34]. The BRG mice with human Sirpα (SRG) were also generated to improve the engraftment efficiency of PDX (Figure 1) [35].

## 3. Establishment and Application of Nude/Hairless Immunocompromised Mice

Although more combined immunocompromised mice have been developed, nude mice remained valuable for human tumor engraftment due to the fact of their advantages for tumor monitoring. Tumor visualization by direct observation and imaging in nude mice provides an excellent opportunity for tumor observation in vivo. For this reason, BALB/c Nude Rag-2/Jak3 double-deficient (Nude R/J) mice were established in our lab by mating nude mice with Rag-2^null^ and Jak3^null^ mice with a BALB/c background [36,37]. Nude R/J mice have no B and T lymphocytes with Rag-2 deficiency, no NK cells with Jak3 deficiency, and have the “don’t eat me signal” with a BALB/c background. Nude R/J mice retained the advantages of no fur and a higher immunocompromisation level than Nude mice, and were, consequently, optimal for in vivo imaging (Figure 2). Another type of furless mouse, namely, the hairless mouse, is also available. Hairless mice have no noticeable immunocompromised phenotypes [38,39]. The SCID Hairless (SHO) mice (Charles River, MA, USA) and Hairless NOD/SCID mice (Envigo, Huntingdon, UK) were established by crossbreeding with hairless mice and used for in vivo imaging (Table 3) [40,41]; however, the engraftment efficiencies were lower than for NK-deficient strains [42]. Mice expressing fluorescence proteins are a powerful tool in cancer research, particularly for visualization of the tumor–host interaction [43]. Several fluorescence-expressing immunocompromised mice have been established and utilized (Figure 3) [44,45,46,47]. The relationships between human tumors and the host microenvironment, including vessels, tumor-associated macrophages (TAMs), and cancer-associated fibroblasts (CAFs), can potentially be studied using these models [48].

## 4. Establishment of PDX Models Using Various Immunocompromised Mice

The PDX models were generated by direct transplantation of patient tumor samples into immunocompromised mice (Figure 4). A significant advantage of PDX models is that they retain key characteristics of the patient’s tumor, such as the gene expression profile and heterogeneity of tumor cells [49,50,51]. Currently, PDX models are the most clinically relevant in vivo cancer models and the most concordant drug response model to human cancer [31,49,52,53,54,55,56]. Thus, the US National Cancer Institute (NCI) decided to substitute a panel of 60 human cell lines (NCI-60) with PDX models for drug screening [3]. 

The duration of tumor establishment in mice differs among tumors, taking from a few days to a few months for the tumor nodule to be observed (first generation; F0). After serial transplantation, the duration of tumor growth becomes stable, with approximately 40–50 days required to obtain certain sized tumors [53,57]. The PDX samples should be stored together with the patient’s clinical data, gene-expression patterns, mutational statuses, drug responsiveness, and pathological analysis to generate a PDX library.

Nude mice have been used to generate PDX models with reasonable efficacy and are used as a standard recipient (Table 4). With this model, the engraftment efficiencies of gastrointestinal tumors are relatively high, while the establishment of hematological malignancy PDXs are almost impossible in nude mice. The introduction of SCID and NOD/SCID mice increased PDX success rates [58]. As NOD/SCID mice have relatively short lifespans and spontaneously develop thymic lymphoma [18], more immunocompromised mice, such as NOG/NSG mice [59,60,61], are a more appropriate model. The NOG/NSG mice are the most immunocompromised mice available and show the highest engraftment efficiencies for both normal and malignant human tissues [53]. However, NOG/NSG mice must be kept under especially clean specific pathogen-free (SPF) conditions; hence, culture prices are relatively high (100 US$ per mouse for academic use in Japan). Moreover, breeding of these mice is not possible by the users. Since nude mice have benefits such as a relatively high engraftment ratios of gastrointestinal tumors, easy observation of subcutaneous tumors, and relatively low price, they remain an important resource for PDX establishment [31,53,62]. The BRJ mice have been used as alternative recipients of cholangiocarcinoma PDX, with a high engraftment ratio (75%) [57,63]. Other solid tumors, such as head and neck tumors, gastric cancers, and bladder cancers, are now under investigation. From our preliminary study, BRJ successfully acquired human solid cancers with relatively high engraftment ratios (compared to currently available models, data not shown) (Figure 5). Since BRJ mice are easy to breed and maintain by users, they are good candidates for PDX, and since Nude R/J mice have the benefits of both BRJ and nude mice, they may be the ideal model for passaging and drug evaluation [36].

Success rates of PDX establishment vary by tumor origin and disease characteristics such as tumor aggressiveness, relapse/recurrence status, and primary or metastatic tumor. More aggressive, relapsed and highly metastatic tumors tend to show higher transplantation rates [53]. Gastrointestinal cancers, such as colon and pancreatic cancers, seem to have higher engraftment ratios compared with other cancers. Engraftment ratios are also higher in more immunocompromised mice (Nude < SCID < NOD/SCID < NSG) (Table 5) [53]. The engraftment ratio for breast cancer is relatively low, and orthotropic transplantation is needed [64]. Orthotropic and renal capsule engraftment clearly increase the engraftment ratio for some tumors, although special techniques are required [31,65,66]. For hematological malignancies such as leukemia and multiple myeloma, direct engraftment into the bloodstream or into the bone marrow of NOG/NSG mice is necessary. Human hormone replacement supports hormone-dependent tumors such as breast and prostate cancers [67,68,69]. 

## 5. Generation of PDX-Derived Cell Lines

A tumor cell line can be generated from a PDX tissue sample [57,70,71]. The establishment of tumor cell lines from primary tissues is relatively difficult with the conventional protocol because fibroblasts often outgrow and overcome the cancer cell growth in vitro. In PDX tissue, human fibroblasts are substituted by murine fibroblasts. These mouse fibroblasts have shorter lifespans and are more sensitive to mechanical and enzymatic removal, so the elimination of fibroblasts takes less time. As mentioned earlier, fluorescence-emitting immunocompromised mice have also been developed. These mice have superior benefits for distinguishing engrafted tumor cells and mouse-derived cells [46,72] and, thus, may be useful for establishing PDX-derived cancer cell lines. These PDX-derived tumor cell lines are useful for high-throughput drug screening, as they retain the characteristics of the primary tumors. It is worth mentioning that, in some cases, male-derived tumor tissues retain the Y chromosome in PDX but lose it during cell line development. This might imply that at least one more mutation is required to establish PDX-derived cell lines [57]. 

## 6. PDX in Humanized Mice

The immune system plays an essential role in tumor control. Recently, cancer immunotherapy using various approaches including antibodies, cancer vaccines, adoptive cell therapies, and immune checkpoint blockade therapies has gained attention as a promising and effective modality with fewer side effects [73,74,75]. However, a mouse model that allows monitoring of the immune response is needed to test these newly developed therapies, because the current PDX model lacks principal immune cells. Mice with a reconstituted human immune system, so-called humanized mice, are available. Humanized mice are generated by transplantation of human hematopoietic stem cells into highly immunocompromised mice such as NOG, NSG, and NOJ [21,22,23,76]. Originally used for pathological studies of human-specific pathogens, these models can mimic the human immune system to a certain degree, but they do not represent a complete and functional human immune system. The mouse bone marrow and thymic microenvironments are different from humans and, therefore, the T cells are not fully developed. Moreover, myeloid and erythroid cell development in these mice are lower than in humans. The humanized bone marrow–liver–thymus (BLT) mouse model can be generated by engraftment of human fetal liver and thymus along with human hematopoietic stem cells into an immunocompromised mouse renal capsule [77]. Since the BTL mouse harbors a nearly complete human immune system including functional T cell response, it is a powerful tool to study human immunology and immunotherapy. However, BLT usage is greatly limited by ethical issues including a restricted supply of the human fetal thymus and liver tissues needed to generate these mice. Human leukocyte antigen (HLA) class I and class II transgenic mice, and several types of NOG/NSG mice with human cytokine transgenes, have been developed to overcome some of these limitations [78,79] (Figure 6). The NSG mice that express human stem cell factor, granulocyte-macrophage colony stimulating factor, and interleukin-3, termed NSG-SGM3, showed robust human hematopoietic reconstitution, a higher frequency of human myeloid cells, and increased regulatory T-cell development [80]. The HLA-expressing humanized mice develop functional HLA-restricted T cells [81,82]. Attempts have also been made to introduce the human hematopoietic microenvironment into immunocompromised mice [83]. These mice represent a very useful model to reconstitute a more accurate human immune system against human malignancies. 

The PDX in humanized mice (humanized PDX) has been established for several types of tumors [91,118,119,120]. The humanized PDX mice model offers a unique platform for examining human acquired and innate immune responses to clinically-relevant tumors and for evaluating immune therapy [121]. However, current models of humanized PDX mice still have several limitations: 1) the balance of hematopoietic and immune cells remains different from that in humans; 2) since the source of HSC is not the same as that of the tumor, it is challenging to match the HLA. These issues require attention for the development of a patient-similar immune response PDX model. 

A specific population of immune cells can be reconstituted in highly immunocompromised mice. Human mature T lymphocytes can reconstitute in the classical PBMC transplanted model, although the duration is relatively short (4–8 weeks) and most of the T lymphocytes are activated [11]. We succeeded in reconstituting human B and T cells and immune response by transplantation of PBMCs into the spleens of NOJ mice [122]. Functional human NK cells and γδT cells can be reconstituted in severe immunocompromised mice and have been used to evaluate the anti-tumor effects of these cells [123,124]. These systems can be applied for the evaluation of cancer immunotherapies such as adoptive cell therapy (NK, gamma delta T (γδT), NK T (NKT), chimeric antigen receptor T (CAR T) cell therapies), antibody therapy (direct killing activity, antibody-dependent cellular cytotoxicity (ADCC), antibody-dependent cellular phagocytosis (ADCP), and complement-dependent cytotoxicity (CDC)), and immune check point blockade therapy. 

## 7. Perspective

Developments in xenograft technology and highly immunocompromised mice such as NOG/NSG allow us to broaden the application of the PDX platform. Nevertheless, PDX models still require optimization for clinical relevance. The human stromal components are rapidly lost and replaced by the murine microenvironment during engraftment [125]. Recently, it was reported that PDX models undergo mouse-specific tumor evolution with rapid accumulation of copy number alterations during PDX passaging that differed from those acquired during tumor evolution in patients by the strong selection pressures in the mice [126], and only selected clones remain after passages. Thus, current PDX models are not complete “Avatar” models of human cancer. In spite of these findings, PDX models are still the most relevant in vivo cancer model for precision medicine, as they keep consistency with their patients’ primary tumor relative to conventional tumor models, especially drug response profiles [127]. Humanized mice with PDX are expected to offer a novel platform for examining immunotherapy (Figure 6) [121]. Despite several limitations, humanized PDX mice have already provided several benefits in studies of cancer behaviors and the functions of immunocompetent cells in tumor microenvironments [118,119,121]. Several attempts have been made to establish humanized microenvironments and generate more comprehensive and functional immune systems in immunocompromised mice [83]. Further development and improvement of these systems will provide an unprecedented platform for personalized cancer medicine, particularly cancer immunotherapy. 

The PDX mice models have emerged as important tools for cancer research, with the potential to allow a personalized approach using gene expression and drug sensitivity profiles. However, they have several limitations that should be noted. Establishment of PDX is time consuming (6 months to 2 years), the success rate varies (10–90%), and it is difficult to retrieve complete patient data. Therefore, many institutions and organizations are focused on creating a large stock of PDX or PDX libraries. European institutions established EurOPDX, a consortium to store PDX, and have already accumulated more than 1500 samples in a PDX bank [128,129]. Jackson Laboratory provides more than 450 samples for researchers [60]. Mega-pharmacies are also establishing their own PDX libraries, and Novartis recently published data on drug screening using 1000 PDX [55]. These PDX biobanks are excellent in vivo platforms for precision medicine [31]. The PDX biobanks with patients’ clinical data, pathologies, gene profiles, and drug response data (Figure 4) are critically important for drug response prediction and validation to generate drug response information for tumors from similar genetic backgrounds. Currently, PDX resources are available in the USA and Europe, and most PDX are derived from common cancers. This might create a bias of information. Hence, PDX biobanks in Asia and rare cancer PDX are essential, as is the requirement for a sharing system among PDX biobanks around the world.

## Figures and Tables

**Figure 1 cells-08-00889-f001:**
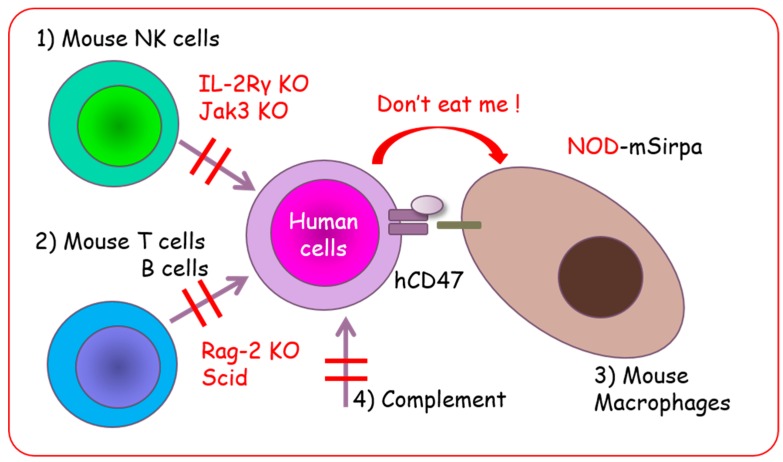
NOG, NSG, and NOJ mice with multiple immune deficiencies are excellent recipients for human cell engraftment. (1) Loss of NK cells; (2) loss of acquired immunity by T and B lymphocyte deficiency; (3) “Don’t eat me” signal by NOD-signal regulatory protein alpha (Sirpα); and (4) loss of complement.

**Figure 2 cells-08-00889-f002:**
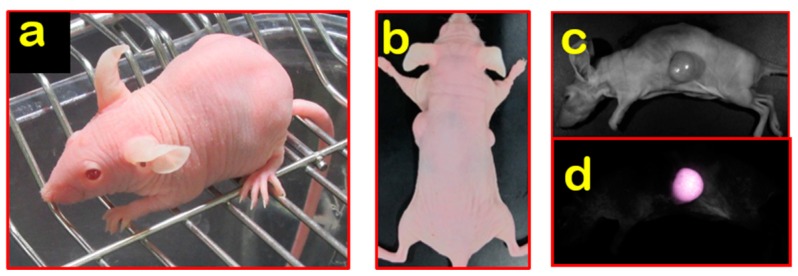
Nude R/J mice: (**a**) BALB/c Nude Rag-2/Jak3 double-deficient (Nude R/J) hairless phenotype; (**b**) direct visualization of subcutaneous tumor nodules in Nude R/J; (**c**–**d**) fluorescent signals observed in Nude R/J.

**Figure 3 cells-08-00889-f003:**
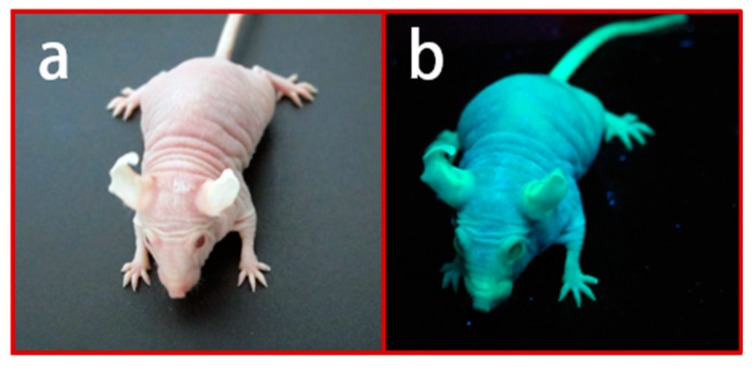
Green fluorescence protein (GFP)-expressing Nude R/J mice: (**a**) GFP Nude R/J phenotype; (**b**) strong GFP expression under β-actin promoter yields a very bright green signal under UV light [46].

**Figure 4 cells-08-00889-f004:**
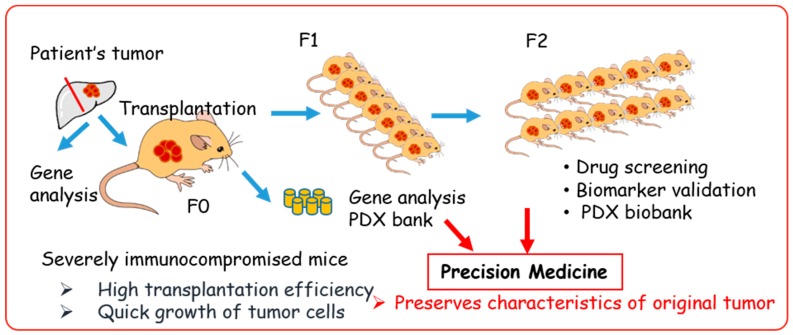
Patient-derived xenograft (PDX) model in precision medicine.

**Figure 5 cells-08-00889-f005:**
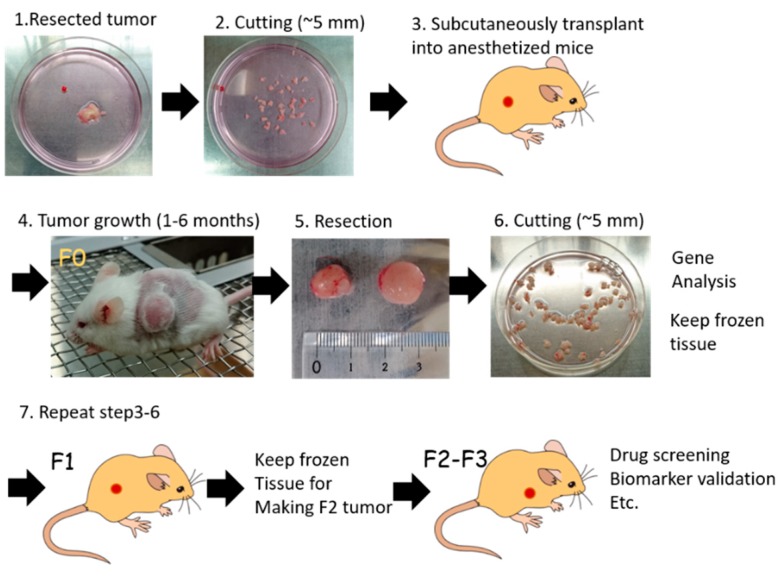
Generation process of PDX. Surgical specimen from a patient’s tumor (1) is divided into small pieces (2) and transplanted into an anesthetized immunocompromised mouse (3). Tumor growth takes 1 to 6 months (4). Once tumors are grown in F0 mice, xenografts are resected (5) and cut into small pieces (6). Parts of tumor tissues are analyzed for tumor characteristics, such as whole exome sequencing (WES), RNA sequencing (RNA-seq), and copy number variation (CNV) analysis. The remnant PDX tumor is stored in liquid nitrogen, or further transplanted into immunocompetent mice (7) for expansion. Conventionally, F2 or F3 PDX tumors are used for cancer biology study, such as drug sensitivity screening, identifying biomarkers, etc.

**Figure 6 cells-08-00889-f006:**
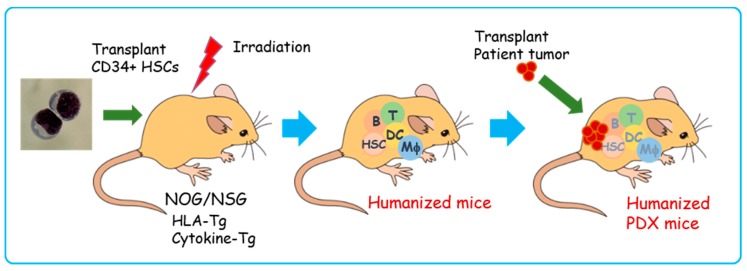
Schematic illustration of humanized PDX model generation. First, CD34+ human hematopoietic stem cells are transplanted into irradiated human leukocyte antigen (HLA)/human cytokine transgenic (Tg) NOG/NSG mice. Then, human hematopoietic and immune systems are reconstituted within 8–12 weeks (humanized mice). Patient-derived tumors are transplanted into humanized mice (humanized PDX mice). T: T lymphocytes, B: B lymphocytes, DCs: dendritic cells, Mφ: macrophages, HSCs: hematopoietic stem cells, HLA-Tg: human leukocyte antigen class I and II transgenic mice, Cytokine-Tg: human cytokine (stem cell factor (SCF), IL-3, granulocyte-monocyte colony stimulating factor (GM-CSF), thrombopoietin (TPO), etc.) transgenic mice.

**Table 1 cells-08-00889-t001:** NOD/SCID-based immunocompromised mice.

Mice	NOD/SCID	NOG	NSG	NOJ
Strain	NOD.Cg-*Prkdc^scid^*	NOD.Cg-*Prkdc^scid^Il2rg^tm1Sug^/Jic*	NOD.Cg-*Prkdc^scid^Il2rg^tm1Wjl^/SzJ*	NOD.Cg-*Prkdc^scid^Jak3^tm1card^*
Genetic defects	SCID	SCID, IL-2γ Partial deficiency	SCID, IL-2Rγ Complete deficiency	SCID, Jak3 deficiency
Developer	CIEA ^1^, Jackson Laboratory	CIEA ^1^	Jackson Laboratory	Kumamoto University
Supplier	Japan Clea Charles River	Japan Clea	Charles River	Kumamoto University
NK activities	NK cell dysfunction	Complete loss of NK cells	Complete loss of NK cells	Complete loss of NK cells
Reference	[18]	[21]	[22]	[23]

^1^ Central Institute for Experimental Animals (CIEA). NOD = non-obese diabetic; SCID = severe combined immunodeficient; NOG = NOD/SCID/IL-2 receptor γ-deficient (IL2Rγ^nul^); NSG = NOD/SCID/IL2Rγ^nul^; NOJ = NOD/SCID/Janus kinase 3 deficient (Jak3); Prkdc = protein kinase, DNA activated, catalytic polypeptide; NK = natural killer.

**Table 2 cells-08-00889-t002:** Comparison of SCID and Rag-1/Rag-2 mutation.

Mice	SCID	Rag-1/Rag-2 Knock Out Mice
Chromosome	Chr.16	Chr.11 p13
Mutated gene	*Prkdc*	Recombination-activation gene-1/-2
Mutation	Natural mutant	Homologous recombination
Immunological phenotype	Deficiency of mature B and T lymphocytes NK cells are normal	Deficiency of mature B and T lymphocytes NK cells were normal
Radiation sensitivity	Sensitive (Lethal dose < 3 Gy)	Normal (Lethal dose 9 Gy)
Leakiness	Leaky	None

Rag = recombination activating gene.

**Table 3 cells-08-00889-t003:** Comparison of hairless mice.

Mice		Hairless	Nude	SCID Hairless	Nude R/J
Strain		BALB/c	BALB/c	CB17.Cg/ICR	BALB/c
Gene abnormality		Hairless	*Foxn1*	Hairless, SCID	*Foxn1*, Rag-2, Jak3
Immune system	T cells	+	−	−	−
	B cells	+	+	−	−
	NK cells	+	+	+	−
Hair coat		None	None	None	None

+: intact certain immune cells, −: lack of certain cells. Foxn1 = forkhead box N1.

**Table 4 cells-08-00889-t004:** Immunocompromised mouse strains for PDX.

Mouse Strain	Phenotype	Advantage	Disadvantage/Consideration	Success Rate of PDX
Nude	No thymus, no coat of hair	Well characterized, easy to detect s.c. tumor	Functional B and NK cells, increased T cell leakage with age	Low
SCID	No mature T and B cells	Better engraftment compared with nude	Functional NK cell, leakage of T cells, radiosensitive	Low
SCID/Beige	No mature T and B cells, impaired Mφ and NK function	Better engraftment compared with SCID	Leakage of T cells, radiosensitive	Moderate
NOD/SCID	No mature T and B cells Impaired NK function Impaired Mφ & DC	Better engraftment	Spontaneous lymphoma Short life span (av. 36wks) Radiosensitive	Moderate
NOG/NSG/NOJ	No mature T and B cells, no NK cells, impaired Mφ and DC	Excellent engraftment of PDX including hematopoietic malignancies	Need strict SPF conditions, breeding is not easy, expensive	High
BALB/c Rag2^null^/IL2Rγ^null^ (BRG) Rag-2 ^null^/Jak3 ^null^ (BRJ)	No mature T and B cells, no NK cells	Excellent engraftment of PDX, resistant to stress, easy breeding, radio resistant		High

NK: natural killer cells, Mφ: macrophages, DCs: dendritic cells, NOG/NSG: NOD/SCID/IL2Rγ^null^, NOJ: NOD/SCID/Jak3null, s.c.: subcutaneous.

**Table 5 cells-08-00889-t005:** PDX success rates in different immunocompromised mice.

Tumor Type	Mice Strain	Implantation Site	Number of Sample	Engraftment Ratio	References
Cholangiocarcinoma	SCID NOD/SCID BRJ	s.c. * s.c. s.c.	55 20 16	34.5% 5.8% 75%	Ojima, 2010 [84] Cavalloni, 2016 [85] Vaeteewoottacharn, 2019 [57]
Colorectal cancer	Nude NOD/SCID NSG	s.c. s.c. s.c	85 85 27	63.5% 87% 54%	Julien, 2012 [86] Bertolini, 2011 [87] Chou, 2013 [88]
Pancreatic cancer	Nude SCID NSG	s.c. s.c. s.c	69 12 121	61% 67% 71.1%	Garrido-Laguna, 2011 [89] Mattie, 2013 [90] Guo, 2019 [91]
Gastric cancer	Nude NOD/SCID Nude/SCID Nude/NOG	s.c. s.c. s.c s.c	32 185 83/119 62	73.7% 34.1% 16.9%/26.9% 24.2%	Wang, 2017 [92] Zhu, 2015 [93] Zhang, 2015 [94] Choi, 2016 [95]
Head and neck cancer	Nude NSG	s.c. s.c.	46 26	54% 84.6%	Keysar, 2013 [96] Kimple, 2013 [97]
Breast cancer	Nude Nude Nude NOD/SCID SCID/Beige NSG	s.c. fat pad ** fat pad fat pad s.c. s.c.	200 314 109 49 162 32	12.5% 2.5% (ER+) 24.3% (ER−) 27% 19% 31.3%	Marangoni, 2007 [98] Cottu, 2012 [99] DeRose, 2011 [100] Zhang, 2013 [101]
Ovarian cancer	Nude Nude SCID SCID NSG	s.c. r.c. *** s.c. s.c. s.c.	138 45 34 168 12	25% 48.8% 50% 74% 83%	Ricci, 2014 [102] Heo, 2017 [103] Dobbin, 2014 [104] Weroha, 2014 [105] Topp, 2014 [106]
Non-small lung cancer	NOD/SCID NOD/SCID NOD/SCID NSG	s.c. r.c. s.c. s.c.	102 527 308 441	25% 90% 26% 28.7%	Fichtner, 2008 [107] Dong, 2010 [108] Chen, 2019 [109] Wang, 2017 [51]
Glioblastoma	NSG	orthotopic	100	30%	Brabetz, 2018 [110]
Prostate	Nude NOD/SCID SCID SCID SCID NSG	s.c. s.c. s.c. orthotopic r.c. s.c.	23 23 86 57 122 27	39% 48% 58.1% 71.9% 93.4% 37%	Priolo, 2010 [111] Wang, 2005 [66] Wetterauer, 2015 [112]
Renal cell carcinoma	Nude NOD/SCID NSG	s.c. r.c. s.c.	336 94 74	8.9% 37.2% 45%	Lang, 2016 [113] Sivanand, 2013 [114] Dong, 2017 [115]
Melanoma	NOG NSG	s.c. s.c.	26 694	88.4% 65.8%	Einarsdottir, 2014 [116] Krepler, 2017 [117]

* s.c., subcutaneous, ** mammarian fat pad, *** r.c., renal capsule.

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
