# Peer review of "Application of Highly Immunocompromised Mice for the Establishment of Patient-Derived Xenograft (PDX) Models"

_cells, 2019, doi:10.3390/cells8080889_

Round 1
Reviewer 1 Report
The manuscript has serious english mistakes. It needs to be carefully written to be understandable. It is not a matter of misspelled words, but wrong sentences, phrases without verbs, phrases that do not make sense, etc
I suggest the authors to review the English language.
Also suggest to present more clear tables.
Author Response
The manuscript has serious english mistakes. It needs to be carefully written to be understandable. It is not a matter of misspelled words, but wrong sentences, phrases without verbs, phrases that do not make sense, etc. I suggest the authors to review the English language. Also suggest to present more clear tables.
[Answer]
Thank you very much for your comments. We asked company for English editing.
Reviewer 2 Report
The authors described an immunocompromised mice for PDX models. This manuscript is interesting and well written.
1. In the table 5, the characteristics of several immunocompromised mice for PDX models were shown. Although the authors mentioned about the engraftment success rate of several cancer types in the table 6, I recommend to add engraftment success rate of PDX models using each strain.
Author Response
The authors described an immunocompromised mice for PDX models. This manuscript is interesting and well written.
1. In the table 5, the characteristics of several immunocompromised mice for PDX models were shown. Although the authors mentioned about the engraftment success rate of several cancer types in the table 6, I recommend to add engraftment success rate of PDX models using each strain.
[Answer]
Thank you very much for the comments. Engraft success rate is better in more severe immunocompromised mice as expected. We added “Success rate of PDX” in Table 5 according to your suggestion.
Reviewer 3 Report
This review provides a useful summary of the different mouse strains used in PDX model generation and the engraftment rates of a small number of different tumour types.
However, in its current form it is certainly not suitable for publication in Cells. The English language is poor, with errors in almost every sentence. It needs extensive editing from someone with an expert grasp on the English language. While the data included is interesting, it only skims the surface and would significantly benefit with a much more comprehensive overview of immunodeficient mouse models (more strains could be included) and tumour engraftment (by including many more tumour types than just six).
The use of references could be significantly improved as there is now a large breadth of literature on PDX tumours that could be cited. For instance, the statement on line 160 has no references, but should cite multiple papers. For line 162, using a single review from 2012 (ref 46) is not appropriate. Multiple research papers should be cited here. I would like to see more recent references too. There are very few references (outside from the authors' own papers) from the last 2-3 years, which in my view are important to include since the PDX field has come a long way in the last 5-10 years.
The use of figures and tables also needs improving. Fig 1 is oversimplified. Some animal strains will have reduced B, T or NK cell function but not diminished, while the impact of the nude mutation is not included. Tables 1-3 and 5 could all be streamlined into a single table that has all the key info and clearly specifies the immunodeficiency status of each mouse strain. There is no Table 4. Table 6 is good, but needs a lot more data. Would be interesting to know the engraftment rates of other tumour types (particularly different types of lung cancer, melanoma, glioblastoma, etc) for which there would be plenty of info in the literature. Would like to see multiple studies per tumour type/mouse strain too rather than just a single study for each.
Other minor comments: Section 2.1 should mention that nude mice can be "leaky" which can result in spontaneous tumour regressions, line 65 needs greater description of Beige mice and the status of NK cells in this strain, while the NSG-SGM3 strain should be included in the humanized mice section
Author Response
This review provides a useful summary of the different mouse strains used in PDX model generation and the engraftment rates of a small number of different tumour types.
However, in its current form it is certainly not suitable for publication in Cells. The English language is poor, with errors in almost every sentence. It needs extensive editing from someone with an expert grasp on the English language. While the data included is interesting, it only skims the surface and would significantly benefit with a much more comprehensive overview of immunodeficient mouse models (more strains could be included) and tumour engraftment (by including many more tumour types than just six).
[Answer]
Thank you very much for the comments. We asked for English editing for revised version.
As the reviewer suggest, there exist more different types of immunocompromised, however; the mice using for PDX are now restricted the strains we show in the Table 5. Moreover, it is going to be more complicated if we add rarely used mice in Table 5. I hope your understanding.
As for the tumor types of PDX in Table 6, we initially intend to show the differences of success rate between mice strain;Nude and Scid are lower success rate compared with NOD/Scid and NOG/NSG, and showed some representative cases. According to the reviewer’s comment, we added more tumor types and adapted one page of Table.
The use of references could be significantly improved as there is now a large breadth of literature on PDX tumours that could be cited. For instance, the statement on line 160 has no references, but should cite multiple papers. For line 162, using a single review from 2012 (ref 46) is not appropriate. Multiple research papers should be cited here. I would like to see more recent references too. There are very few references (outside from the authors' own papers) from the last 2-3 years, which in my view are important to include since the PDX field has come a long way in the last 5-10 years.
[Answer]
Thank you very much for the comments. The statement of line 160 and 162 are based on same reference from 2012. According to the reviewer’s comment, we added several key references. We also added up-to data references of PDX.
The use of figures and tables also needs improving. Fig 1 is oversimplified. Some animal strains will have reduced B, T or NK cell function but not diminished, while the impact of the nude mutation is not included. Tables 1-3 and 5 could all be streamlined into a single table that has all the key info and clearly specifies the immunodeficiency status of each mouse strain. There is no Table 4. Table 6 is good, but needs a lot more data. Would be interesting to know the engraftment rates of other tumour types (particularly different types of lung cancer, melanoma, glioblastoma, etc) for which there would be plenty of info in the literature. Would like to see multiple studies per tumour type/mouse strain too rather than just a single study for each.
[Answer]
Thank you very much for the valuable comments.
Figure 1 is the scheme of only NOG, NSG, and NOJ mice, which has multiple deficiencies to recognize human cells, not included Nude mice and other types of immunocompromised mice. Since NOG/NSG mice are the key mice to establish PDX, we intend this scheme help the understanding of multiple immune deficiency of the mice.
Thank you very much for the comments for Tables. If we can combine Table1-4 and make single table, it can be more sophisticated table. However, we think it will be more complicated and not easy to understand for the readers.
We added more tumor types for Table 5 according to reviewer’s suggestion.
We corrected the numbering the Tables according to reviewer’s comment.
Other minor comments: Section 2.1 should mention that nude mice can be "leaky" which can result in spontaneous tumour regressions, line 65 needs greater description of Beige mice and the status of NK cells in this strain, while the NSG-SGM3 strain should be included in the humanized mice section
[Answer]
Thank you very much for valuable comments. According to the reviewer’s comments, we added some description of “leaky” of Nude mice, character of Beige mice, and NSG-SGM3 strain.
Reviewer 4 Report
In their manuscript, Okada et al. provide insights into availability and selected applications of immune deficient mouse models for xenotransplantation of human patient tissue. In detail, they provide an overview on the historical development of immune deficient mouse strains with a focus on the subsequent generation of increasingly more immune compromised animals. Moreover, they shortly summarize the usability of PDX as a resource for subsequent cell line generation and describe approaches to create further improved PDX models by introducing a humanized environment into the mouse prior to grafting the patient tissue.
Main comment:
The authors here discuss a timely and relevant technique which is an essential tool for current translational precision oncology applications. However, while the review title suggests a comprehensive overview on clinical application, after describing different available mouse strains for xenografting, the authors restrict this aspect to a tiny paragraph mentioning the usability of xenograft models for cell line generation and then again describe efforts towards generation of more efficient mouse models. To address the title, a more comprehensive discussion of relevant current clinical applications, e.g. xenografts as “living biobanks” for drug testing and others, is surely needed. However, as these topics are extensively reviewed elsewhere, the authors may indeed restrict the review topic on the development and level of immune deficiency of mouse strains for xenografting patient material. If they decide to do so, however, the major problems and challenges of these models (e.g. engraftment of patient derived non-cancerous cells and consequences for potential applications, intra-tumor heterogeneity/ clonal shifts, and others) need to be raised and discussed in more detail. Alternatively, as the review is part of a special issue relating to cancer models, the authors may potentially consider to discuss and review the topics raised by the accompanying original research papers in more depth.
Additional comments:
- The paragraph showing generation process of subcutaneous xenografts in my opion does not add to gain an overview of xenograft models for clinical applications and should be deleted
- The manuscript requires extensive language editing
- Summarizing figures need to be professionalized
Author Response
In their manuscript, Okada et al. provide insights into availability and selected applications of immune deficient mouse models for xenotransplantation of human patient tissue. In detail, they provide an overview on the historical development of immune deficient mouse strains with a focus on the subsequent generation of increasingly more immune compromised animals. Moreover, they shortly summarize the usability of PDX as a resource for subsequent cell line generation and describe approaches to create further improved PDX models by introducing a humanized environment into the mouse prior to grafting the patient tissue.
Main comment:
The authors here discuss a timely and relevant technique which is an essential tool for current translational precision oncology applications. However, while the review title suggests a comprehensive overview on clinical application, after describing different available mouse strains for xenografting, the authors restrict this aspect to a tiny paragraph mentioning the usability of xenograft models for cell line generation and then again describe efforts towards generation of more efficient mouse models. To address the title, a more comprehensive discussion of relevant current clinical applications, e.g. xenografts as “living biobanks” for drug testing and others, is surely needed. However, as these topics are extensively reviewed elsewhere, the authors may indeed restrict the review topic on the development and level of immune deficiency of mouse strains for xenografting patient material. If they decide to do so, however, the major problems and challenges of these models (e.g. engraftment of patient derived non-cancerous cells and consequences for potential applications, intra-tumor heterogeneity/ clonal shifts, and others) need to be raised and discussed in more detail. Alternatively, as the review is part of a special issue relating to cancer models, the authors may potentially consider to discuss and review the topics raised by the accompanying original research papers in more depth.
[Answer]
Thank you very much for valuable comments. As shown in the title, we focus on the use of immunocompromised mice for PDX. We added some description and discussion for the text.
Additional comments:
-The paragraph showing generation process of subcutaneous xenografts in my opinion does not add to gain an overview of xenograft models for clinical applications and should be deleted.
[Answer]
Thank you very much for the comment. We show this Figure for the readers to imagine the process of making PDX. We hope your understanding.
-The manuscript requires extensive language editing
[Answer]
We asked for English editing for revised version.
-Summarizing figures need to be professionalized
[Answer]
We tried to be easily understandable schemes.
Round 2
Reviewer 3 Report
The editing changes to the English have made a major difference and greatly improved the readability of the manuscript. However, I don't think the edits to the title are appropriate. A better title in my view would be: "Application of highly immunocomprimised mice for establishment of patient-derived xenograft (PDX) models"
Table 5 (previously table 6) is vastly improved although could still benefit by including sample size rather than just %. (ie if engraftment rate is 75% but only 3 of 4 models have grown than the data is fairly meaningless, but if its 75 out of 100 then it holds much more weight)
The addition of more references has helped too, although there is still no mention of the Ben-David et al. 2017 paper in Nature Genetics (doi:10.1038/ng.3967) that provides a cautionary view on the benefit of PDXs and would provide a bit more balance to the manuscript. This should be added to the limitations paragraph in the perspectives section, with a bit more detail of the scientific limitations of PDXs not just the experimental limitations
Author Response
The editing changes to the English have made a major difference and greatly improved the readability of the manuscript. However, I don't think the edits to the title are appropriate. A better title in my view would be: "Application of highly immunocomprimised mice for establishment of patient-derived xenograft (PDX) models"
[Answer]
Thank you very much for the comments. According to the reviewer’s comment, we changed the title of our manuscript.
Actually, English editor changed our initial title. We also feel the title of reviewer’s recommendation is appropriate.
Table 5 (previously table 6) is vastly improved although could still benefit by including sample size rather than just %. (ie if engraftment rate is 75% but only 3 of 4 models have grown than the data is fairly meaningless, but if its 75 out of 100 then it holds much more weight)
[Answer]
Thank you very much for the comment. We completely agree the reviewer’s opinion. We mostly chose the references based on relatively large number of samples. However, as sample numbers are different from tumor type and it has benefit for the readers, we added “number of samples” in Table 5.
The addition of more references has helped too, although there is still no mention of the Ben-David et al. 2017 paper in Nature Genetics (doi:10.1038/ng.3967) that provides a cautionary view on the benefit of PDXs and would provide a bit more balance to the manuscript. This should be added to the limitations paragraph in the perspectives section, with a bit more detail of the scientific limitations of PDXs not just the experimental limitations
[Answer]
Thank you very much for your valuable comment. We added some discussion based on this paper in the perspective.
Reviewer 4 Report
The authors made a good effort to address this reviewers' concerns.
Author Response
Thank you very much for your kind comments and suggestions.